# Sex Differences in the Brain Transcriptomes of Adult Blue Gourami Fish (*Trichogaster trichopterus*)

Gad Degani [1,2] and Ari Meerson [1,2,*]

1   MIGAL Galilee Research Institute, Tarshish 2, Kiryat Shmona 1101602, Israel
2   Faculty of Sciences, Tel-Hai Academic College, Upper Galilee 1220800, Israel
*   Correspondence: arim@migal.org.il; Tel.: +97-2546320771

**Abstract:** Blue gourami (gourami, *Trichogaster trichopterus*) is a model for labyrinth fishes (Anabantoidei) adapted to partial air breathing. Its reproductive endocrinology has been extensively studied, and transcriptomic sex differences in the gonads were described. Nevertheless, sex differences in gene expression in non-gonadal tissues ostensibly affected by the sex-specific hormonal balance, e.g., the brain, are unknown. To assess such differences, we used bulk RNA-seq to assemble and compare polyA+ transcriptomes between whole brains of four adult male and five adult female gourami, in addition to other tissues (three dorsal fin and five ovary samples) from the same female group. While all nine brain transcriptomes clustered together relative to the other tissues, they showed separation according to sex. A total of 3568 genes were differentially expressed between male and female brains; of these, 1962 and 1606 showed lower and higher expression in males, respectively. Male brains showed stronger down-regulation of specific genes, which included hormone receptors, e.g., pituitary adenylate cyclase-activating polypeptide receptor (*pacap-r1*). Among the genes with lower expression in male brains, multiple pathways essential to brain function were over-represented, including GABA, acetylcholine and glutamate receptor signaling, calcium and potassium transmembrane transport, and neurogenesis. In contrast, genes with higher expression in male brains showed no significant over-representation of brain-specific functions. To measure the mRNA levels of specific hormone receptors known from prior studies to regulate reproductive function and behavior in gourami and to validate RNA-seq results for these specific genes, we performed RT-qPCR for five receptors, *pacap-r1*, gonadotropin-releasing hormone 2 receptor (*gnrh2r*), kisspeptin receptor 1 (*gpαr1/kiss1*), insulin-like growth factor 1 receptor (*igf1r*), and membrane progesterone receptor 1 (*mpr1*), in the brain RNA sample groups. Of these, *pacap-r1* showed a significant, three-fold down-regulation, while *gpαr1/kiss1* showed a significant two-fold down-regulation in male vs. female gourami brains. Our results are novel in describing the suppression of brain function-related gene expression in male, as compared to female, gourami brains. Further research is needed to assess the behavioral significance of this effect and its prevalence in other vertebrate groups.

**Keywords:** labyrinth fishes; Anabantoidei; sexual dimorphism; neural function; gene expression; transcriptomics

**Key Contribution:** Although sex-based differences in brain gene expression are widely studied, to our knowledge our results are novel in describing the suppression of neural function in adult gourami males as compared to females. Further research is needed to assess the behavioral significance of this effect and its prevalence in other vertebrate groups.

## 1. Introduction

The brain controls central signaling pathways affecting the differences between males and females in fish (see review [1]). The differences between male and female *Danio rerio* (zebrafish) brains are studied to understand their differentiation and as a model for

sex-specific neurological diseases in humans. Male and female brains display subtle differences at the cellular level, which may be important in driving sex-specific signaling, behavior, and responses to neuropharmacological treatments [2]. Thus, aggressive tendencies, apprehension, nervousness, and schooling behaviors show distinct differences between males and females in zebrafish [3]. Differences in brain metabolites between male and female zebrafish may underlie behavioral variations between the sexes. These distinctions are observed across species, hinting at a universal phenomenon. Although traditional theories attribute brain sex differences primarily to gonadal steroid hormones, recent findings suggest that genetic disparities play a role even before hormonal influence begins [1]. Exploring the intricate signaling pathways involved in brain sex differentiation holds promise for understanding sex-specific neurological conditions [4,5]. The zebrafish model greatly facilitated this field of research, further enhanced by RNA sequencing. Intersex differences extend to cellular and metabolic levels, potentially shaping sex-specific neuronal communication [1]. Furthermore, the genetic similarity of zebrafish to humans, alongside structural parallels, offers a valuable platform for unraveling brain-associated sex differentiation and its implications for diagnosing and treating neurological disorders [1].

The blue gourami (gourami, *Trichogaster trichopterus*) belongs to the suborder Anabantoidei in the order Perciformes. This small tropical freshwater fish possesses a unique feature—a labyrinth chamber above its gills—that allows it to extract oxygen from the air for respiration. This adaptation enables the blue gourami to thrive in aquatic environments with low dissolved oxygen levels by facilitating partial air breathing. Anabantoid fishes are primarily found in central Africa, India, and southern Asia [6–10]. In their natural habitats, they demonstrate remarkable adaptability to fluctuating dissolved oxygen levels in the water throughout the year, which can sometimes plummet to extremely low levels [9,11]. Approximately 80 species are distributed among the 16 recognized genera of Anabantoid fishes [11]. However, the precise classification of labyrinth fishes remains disputed among taxonomists, leading to the use of numerous synonyms. Reference [11] suggests that labyrinth fishes can be classified into four families, one of them being the Anabantidae.

Due to the number of genes functionally characterized in gourami, it is considered a model labyrinth fish. These genes, many of which play key roles in the processes of growth and reproduction, have been analyzed as indicators of divergence between species [12,13]; they include genes responsible for the production of several hormones, such as kisspeptins 1 and 2 [14,15]; gonadotropin-releasing hormones (*gnrhs*) 1, 2, and 3 [16,17]; growth hormone (*gh*) [8,15,18]; somatolactin [12]; prolactin (*prl*) [19]; follicle-stimulating hormone (*fsh*); luteinizing hormone (*lh*) [20]; and mitochondrial genes encoding cytochrome b and 12S rRNA [21]. Notably, the genetic markers in gourami show differences compared to other bony fish. Researchers frequently rely on the sequence of the mitochondrial cytochrome c oxidase subunit 1 gene to conduct studies on the Anabantidae. Furthermore, among the genes that regulate growth and reproduction, the most effective genetic markers for distinguishing between species of the Anabantoidei appear to be those associated with the functioning of the hypothalamic–pituitary–somatotropic (HPS) axis, namely, pituitary adenylate cyclase-activating polypeptide (*pacap*) and its related peptide *prp* (formerly known as *ghrh*-like peptide), in addition to the 12S rRNA gene [12,21]. However, transcriptomic information on Anabantidae is limited.

A whole-brain transcriptome study of female gourami showed changes in the expression of genes during the process of vitellogenesis [22], including those encoding reproductive hormones. These hormones include kisspeptin 2, its receptors, *kissr1* and *kissr2*, and *gnrh1*, *gnrh2*, and *gnrh3*, along with their corresponding receptors. In addition, the hormones *pacap* and *prp*, somatolactin, *fsh*, *lh*, growth hormone (*gh*), *prl*, estradiol (*e2*), testosterone, vitellogenin, and 17α,20β-dihydroxy-4-pregnen-3-one (*17, 20p*) are involved in gourami reproduction [4]. Nevertheless, sex differences in gene expression in non-gonadal tissues ostensibly affected by the sex-specific hormonal balance, e.g., the brain, are unknown; assessing such differences was the focus of the current study.

## 2. Materials and Methods

### 2.1. Fish and Sampling Procedure

Male and female blue gourami (gourami, *T. trichopterus*) were bred and maintained at a fish farm located at Moshav Shorashim in the Misgav area, northern Israel. Investigations were conducted under the supervision of an in-house veterinarian, in accordance with the AVMA Guidelines for the Euthanasia of Animals, American Veterinary Medical Association [23]. The fish were raised in containers with dimensions of $2 \times 2 \times 0.5$ m$^3$ at a temperature of 27 °C, under a 12 h light/12 h dark photoperiod [24]. The fish were fed an artificial diet with a protein content of 45% and a fat content of 7% [25]. Each fish was anesthetized in a clove oil bath (0.25 mg/L), and its weight and length were recorded. Average mean body weight (BW) of the mature females was $6.52 \pm 0.61$ g [26] and that of the males was $8.0 \pm 2.6$ g [7]. Brain samples were collected from mature males and females as described previously [14,25]. Gonad samples (ovary, $N = 5$) and dorsal fin samples ($N = 3$) were taken from the same female gourami as the brain samples, in order to serve as possible out-groups in clustering analysis of the brain samples.

### 2.2. RNA Extraction

RNA was extracted as described previously [22]. Briefly, samples were kept in RNALater (Thermo-Fisher Scientific, Waltham, MA, USA) following excision and homogenized using a TissueRuptor (Qiagen, Hilden, Germany). Whole RNA from each sample was extracted using TRI Reagent (Sigma, Kawasaki, Japan), following the manufacturer's guidelines. The concentration and integrity of the RNA were assessed using a Thermo-Fisher Scientific NanoDrop 8000 spectrophotometer and Agilent TapeStation 4150. All RNA samples included in this study exhibited OD$_{260/280}$ > 1.8 and RIN > 7.

### 2.3. Library Prep and Sequencing

RNA-seq libraries were prepared at the Crown Genomics Institute of the Nancy and Stephen Grand Israel National Center for Personalized Medicine, Weizmann Institute of Science (Rehovot, Israel), using the in-house polyA-based RNA seq protocol (INCPM mRNA-Seq). Briefly, the polyA fraction (mRNA) was purified from 500 ng of total input RNA followed by fragmentation and generation of double-stranded cDNA. After cleanup with Agencourt AMPure XP beads (Beckman Coulter, Brea, CA, USA), end repair, A base addition, adapter ligation, and PCR amplification steps were performed. Libraries were quantified by Qubit (Thermo Fisher Scientific, Waltham, MA, USA) and TapeStation (Agilent, Santa Clara, CA, USA). Sequencing libraries were constructed with barcodes to allow multiplexing on an Illumina NovaSeq 6000 machine, using an SP (100 cycles) kit. 100bp single reads were sequenced on two lanes. Fastq files for each sample were generated by bcl2fastq v2.20.0.422 (Illumina, San Diego, CA, USA). Poly-A/T stretches and Illumina adapters were trimmed from the reads using cutadapt; resulting reads shorter than 30bp were discarded.

### 2.4. Transcriptome Assembly

The trimmed reads were subsequently employed for assembly purposes using Trinity (v2.13.2) [27]. Post-assembly, read representation was evaluated using Bowtie2 (v2.3.4.3) [28], and completeness was assessed via BUSCO (v5.4.4) [29]. The process of read clustering was carried out using CD-HIT (v4.8.1) [30]. Trans Decoder (v5.7.0) (Haas, BJ. https://github.com/TransDecoder/TransDecoder (accessed on 16 July 2024)) was used to predict the coding regions in each gene's longest transcript. Functional annotations were added to the protein sequences using eggNOG mapper [31]. The resulting annotation was merged with the differential expression analysis results (below).

### 2.5. Differential Expression (DE) Analysis

Abundance estimation was performed by RSEM (alignment-based) [32]. The resulting counts matrix (Trinity_genes.isoform.counts.matrix) was used for DE analysis. Differen-

tially expressed genes were identified using DESeq2 [33] with the betaPrior, cooksCutoff and independentFiltering parameters set to False. Default DESeq2 normalization was performed to adjust for library read counts. Raw *p* values were adjusted for multiple testing using the procedure of Benjamini and Hochberg [34]. The pipeline was run using snakemake [35]. The 1000 most variable genes from the DESeq2 analysis were used for Principal Component Analysis (PCA) and hierarchical clustering (HCL) plots using default values. Volcano plots were prepared using the ggplot R package using DESeq2 output.

### 2.6. Pathway Analysis

Functional enrichment analysis was performed using g:Profiler (version e109_eg56_p17 _1d3191d) with the g:SCS multiple testing correction method, applying a significance threshold of 0.05 [36]. *Danio rerio* was used as a reference species for the gene lists. The gene lists contained DE transcripts, either up- or down-regulated in each comparison, with adjusted *p*-values < 0.05.

A schematic of the study's analytical pipeline appears in Figure 1.

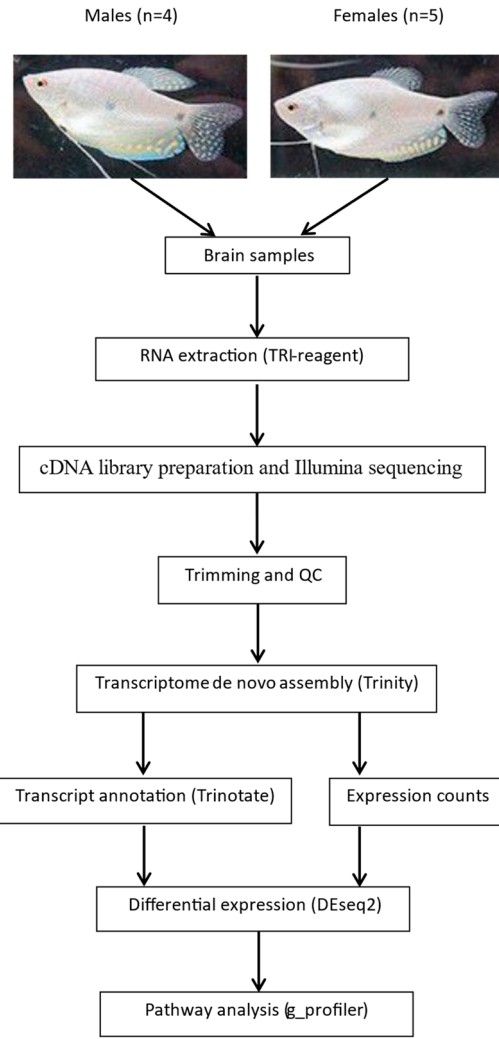

**Figure 1.** Scheme of experimental design and analysis.

### 2.7. RT-qPCR

Verso cDNA Synthesis Kit (ThermoFisher Scientific, Waltham, MA, USA) was employed to synthesize cDNA for RT-qPCR from the RNA extracted above (Section 2.2) based on the manufacturer's recommendations. Briefly, RNA (0.5 μg) was heated to 70 °C for 5 min and cooled to 4 °C before adding the anchored-oligo dT primer and other reagents

supplied in the kit, including the RT enhancer that prevents genomic DNA carryover. The cDNA synthesis reaction was performed using a BioEr GeneExplorer PCR cycler at 42 °C for 60 min, followed by 52 °C for 30 min, and enzyme inactivation at 95 °C for 2 min. Quantitative PCR was performed using SYBR Green chemistry and DNA primers from Sigma, Israel (sequences are provided in Supplementary File S5). All primers were tested for efficiency (by serial dilutions) and specificity (by melting peak analysis). qPCR was performed in technical quadruplicates on a BioRad CFX-384 qPCR device. Data were analyzed using CFX Maestro software v.2.3 (BioRad, Hercules, CA, USA) and Microsoft Excel. Relative quantification and the ΔCq method were used. Results were normalized to input RNA.

## 3. Results

### 3.1. Brain Transcriptomes Tend to Cluster by Sex

To evaluate sex-associated differences in gene expression in the gourami brain, as a non-gonadal tissue ostensibly affected by the sex-specific hormonal balance, we used bulk RNA-seq to assemble and compare polyA+ transcriptomes between whole brains of four adult male and five adult female gourami, in addition to three dorsal fin and five ovary samples from the same female group. Between 14.7 and 40.6 million reads were obtained per sample. Trimmed reads from all fish and all tissues sampled were used for transcriptome assembly, resulting in 238,913 contigs clustered into 168,882 "genes". The alignment rate of reads from individual samples to the assembly was between 95.1 and 97.8%. The evaluation of completeness based on conserved ortholog content produced a score of 98.8%: 252 complete (of which 90 were single-copy and 162 were duplicated), 3 fragmented, and 0 missing reads among 255 total BUSCO groups searched.

To compare the effects of different factors on the overall gene expression profile, and for detecting technical batch effects (if any) and outliers, samples were clustered by gene-expression profile, based on the 1000 most variable genes. While all nine brain transcriptomes clustered together relative to the other tissues regardless of sex (Figure 2A), they showed separation according to sex (Figure 2B). Hierarchical clustering of brain samples based on gene expression showed a similar, almost full segregation by sex (Figure 2C).

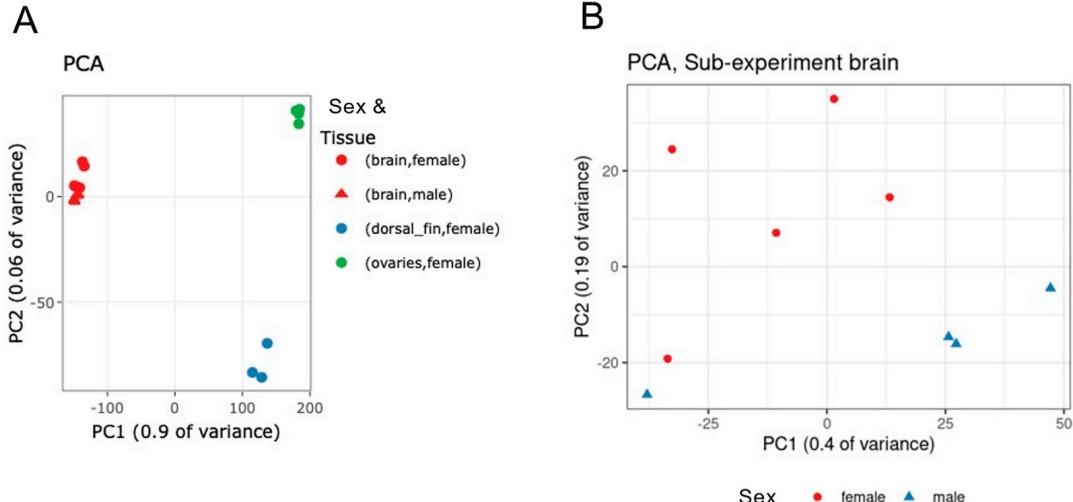

**Figure 2.** *Cont.*

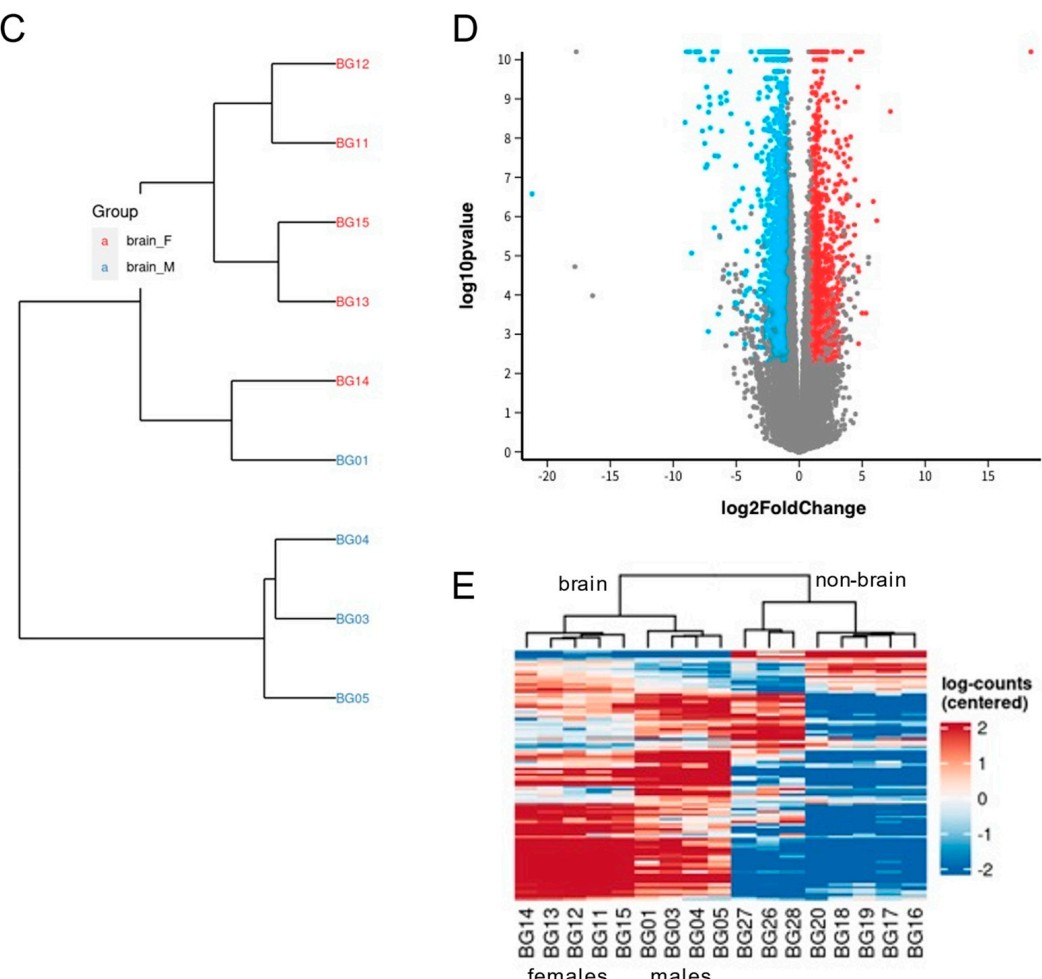

**Figure 2.** Adult gourami brain transcriptomes tend to cluster by sex. (**A**) PCA of all tissues; (**B**) PCA of brain samples; (**C**) hierarchical clustering (HCL) of brain samples; (**D**) volcano plot of brain DEseq2 results (male vs. female, significantly up- or down-regulated genes in red and blue, respectively); (**E**) heatmap of significantly DE genes from DEseq2 results (all tissues).

### 3.2. Similar Numbers, but Not Magnitude of Expression Difference, of Genes Up- and Down-Regulated in Males vs. Females

We used DESeq2 [33] to identify genes differentially expressed (DE) between male and female brains. A total of 3568 DE genes were identified; of these, 1962 and 1606 showed lower and higher expression in the males, respectively. The full list of DE genes can be found in Supplementary File S3. The particularly strong down-regulation of some genes in the male brains was apparent from the asymmetry of the volcano plot of DE genes (Figure 2D). Hierarchical clustering and heatmaps of gene expression based on genes identified by DESeq2 showed full segregation of brain samples by sex, as well as the segregation of brain samples from other tissues (Figure 2E).

### 3.3. Multiple Pathways Essential to Brain Function Are Over-Represented among Genes Down-Regulated in Males vs. Females, but Not Vice Versa

To assess the functional profile of DE genes, we used g.profiler [35] with lists of genes either up- or down-regulated in male vs. female brains. Among the genes with lower expression in male brains, multiple pathways essential to brain function were over-represented, including the GABA, acetylcholine and glutamate receptor signaling, calcium and potassium transmembrane transport, and neurogenesis pathways (Figure 3A). In contrast, the genes with higher expression in male brains showed no significant over-representation of brain-specific functions; additionally, the confidence scores of the over-

represented functions are dramatically lower (Figure 3B). Full g.profiler results are available in Supplementary File S4.

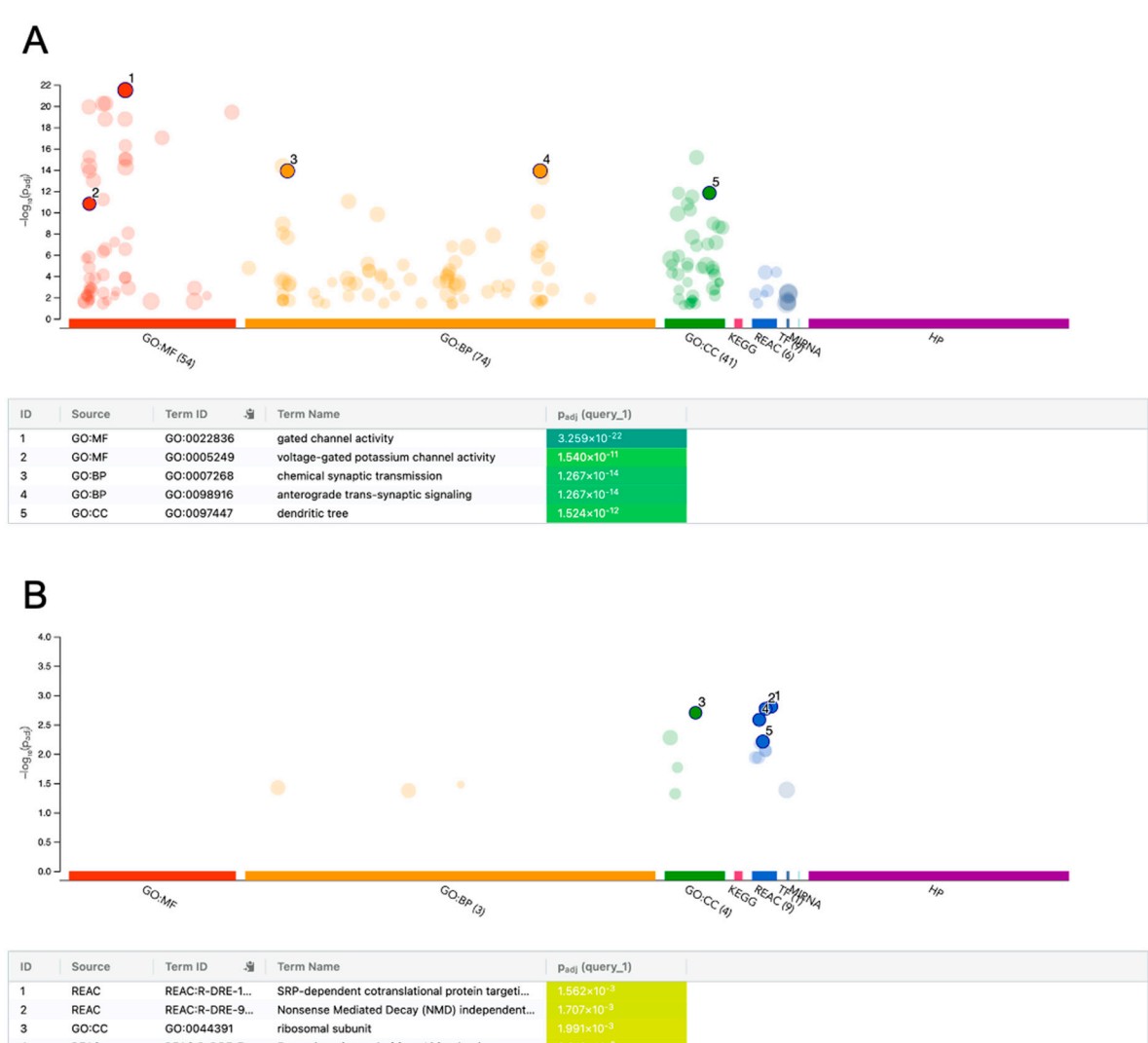

**Figure 3.** G.profiler graphs and compact tables of selected over-represented terms among genes that were (**A**) down-regulated and (**B**) up-regulated in male vs. female gourami brains.

*3.4. Hormone Receptors Involved in Regulation of Reproduction Are Down-Regulated in Male Gourami Brains*

To check the expression of specific hormone receptors known from prior studies to regulate reproductive function and behavior in gourami, and to validate RNA-seq results for specific genes, we performed RT-qPCR on the brain RNA sample groups, using primers for five receptors: pituitary adenylate cyclase-activating polypeptide receptor 1 (*pacap-r1*), gonadotropin-releasing hormone 2 receptor (*gnrh2r*), G protein-coupled receptor 1A/kisspeptin receptor 1 (*gpαr1/kiss1*), insulin-like growth factor 1 receptor (*igf1r*), and membrane progesterone receptor 1 (*mpr1*). The primer sequences are provided in Supplementary File S5. Of the genes thus assayed, *pacap-r1* showed a significant, three-fold down-regulation in male vs. female gourami brains (Figure 4A), which closely mirrored the three-fold down-regulation of *pacap-r1/adcyap1r1* in the RNA-seq dataset (Figure 4B and Supplementary File S3, line 753 in the pass-down worksheet). *gpαr1/kiss1* showed a more modest, yet still significant two-fold down-regulation in the male vs. female gourami brains. *igf1r*, which showed a significant, 2.7-fold down-regulation in male vs. female gourami brains in the RNA-seq results (Figure 4B and Supplementary File S3, line 998

in the pass-down worksheet), showed a non-significant trend of down-regulation in the RT-qPCR results, as did *mpr1* (Figure 4A). *gnrh2r* showed highly variable expression among individual samples (Figure 4A).

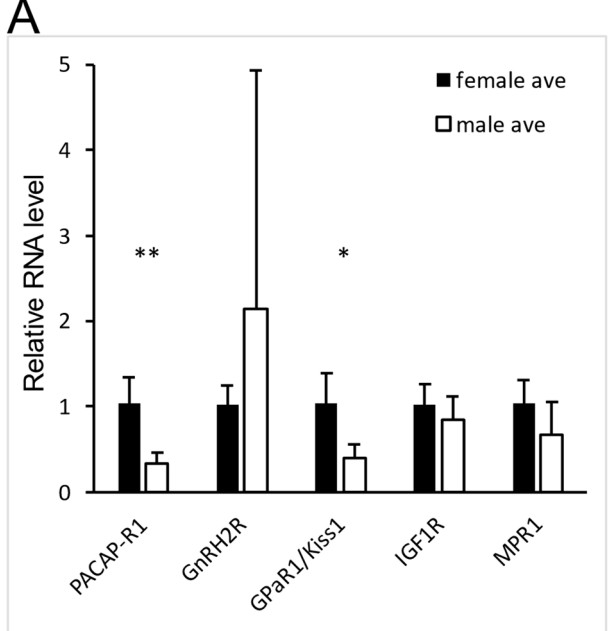 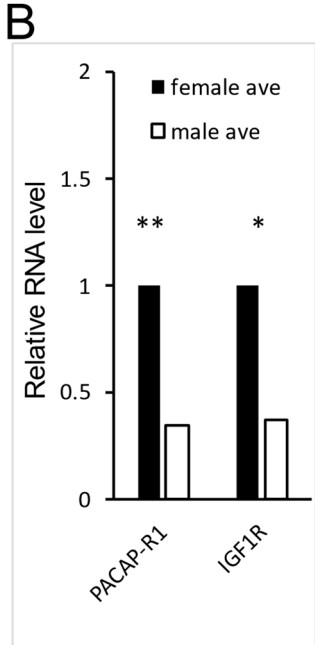

**Figure 4.** A. Relative mean mRNA abundance (based on RT-qPCR), in female and male gourami brains, of hormone receptors as follows: *pacap-r1* (pituitary adenylate cyclase-activating polypeptide receptor 1), *gnrh2r* (gonadotropin-releasing hormone receptor), *gpar1/kiss1* (G protein-coupled receptor 1 (A), *igf1r* (insulin-like growth factor 1 receptor), and *mpr1* (membrane progesterone receptor 1). *: $p = 0.01$; **: $p = 0.0038$ (*t*-test). Bars, *stdev* (N = 5). (**B**): Relative mean mRNA abundance (based on RNA-seq, see Supplementary File S3), in female and male gourami brains, of two of the same hormone receptors (*pacap-r1* and *igf1r*). *: $p = 0.0153$; **: $p = 0.0002$ (adjusted *p*-values, DESeq2).

## 4. Discussion

Our results indicate a broad suppression of brain-function-related gene expression in male, as compared to female, gourami brains. This result agrees with previous findings in zebrafish [1,3], where gene expression patterns differed between male and female brains. These zebrafish studies also showed that male and female brains differed in neuronal structure, namely the size, shape, and connectivity of neurons, which can influence neural circuitry and function. Additionally, they observed sex-based differences in the distribution, abundance, and activity of neurotransmitters such as dopamine, serotonin, and glutamate, which play crucial roles in neural signaling and behavior. Thus, hormonal regulation pathways showed distinctions between male and female zebrafish brains, which is mirrored on the transcriptome level in the present study in gourami. Hormones like testosterone, estrogen, and progesterone exert differential effects on neural development, synaptic plasticity, and behavioral responses in each sex. Male and female zebrafish, as well as gourami [4,22], display contrasting behaviors in line with underlying neurological differences in neural processing, sensory perception, or responses to environmental stimuli. Also, specific brain regions were activated differently in male and female zebrafish [1,3] and gourami [4,22] in response to various stimuli or tasks, suggesting divergent neural processing and integration of sensory information between the sexes.

Sex-based differences in gene expression in the gourami brain likely contribute to differences in downstream hormonal signaling, which have been extensively studied. Thus, in male gourami, *gnrh1* regulates spermatogenesis through the brain–pituitary–gonad (BPG) axis during both non-reproductive and reproductive stages by controlling *fsh*, 11-

ketotestosterone (*11kt*), and 17β-estradiol (*e2*). Conversely, during the reproductive stage, *gnrh3* has a more pronounced impact through the BPG axis (specifically on the transition from producing spermatids to sperm) by modulating *lh*, *11kt*, and 17α-hydroxyprogesterone (*17p*). Simultaneously, the HPS axis plays a role in the nervous system's involvement in spermatogenesis [16]. In female gourami, the process of vitellogenesis within the brain is likewise accompanied by changes in gene expression [22]. The key hormones in the reproductive processes of female gourami include *kiss2*, its receptors, *kissr1* and *kissr2*, and *gnrh1*, *gnrh2*, and *gnrh3*, along with their corresponding receptors. Additionally, the hormones *pacap* and *prp*, somatolactin, *fsh*, *lh*, *gh*, *prl*, *e2*, testosterone, vitellogenin, and 17α, 20β-dihydroxy-4-pregnen-3-one (*17,20p*) play vital roles [4].

We used RT-qPCR to evaluate the expression of selected receptors involved in the aforementioned pathways, as well as to validate the RNA-seq data for those of the receptor genes that were identified as differentially expressed. For example, the *pacap* receptor was strongly down-regulated in male gourami brains, in agreement with our prior findings.

The sex-specific hormonal pathways that act downstream of the gourami brain ostensibly affect the brain itself via feedback loops, as is well known in other vertebrates [37]. This feedback likely contributes to the differences in brain gene expression observed in the current study (scheme, Figure 5).

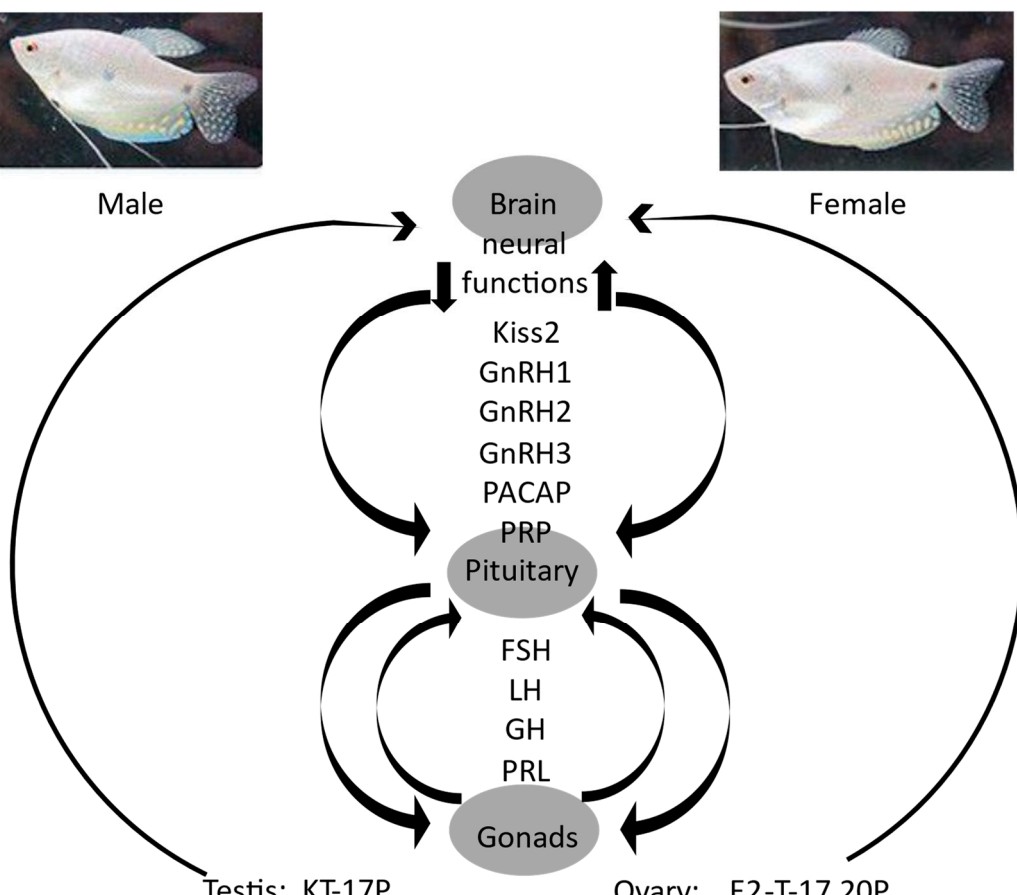

**Figure 5.** Scheme of suggested model—inter-regulation between brain function pathways and sex-specific hormone signaling in gourami. The brain–pituitary–gonad signaling axis in gourami, its principal hormones, and receptors have been extensively described by the first author and colleagues (review, [4]). The feedback of sex hormones affecting the brain is known from zebrafish and other species; the male-specific down-regulation of genes involved in essential brain functions is reported in the current study.

Although sex-based differences in brain gene expression are widely studied, to our knowledge the current results are novel in describing a broad suppression of neural function-related gene expression in adult gourami males as compared to females. Further research is needed to assess the behavioral significance of this effect and its prevalence in other vertebrate groups.

## 5. Conclusions

1. Gene expression in the brains of adult blue gourami differs broadly between the sexes;
2. Brain-function-related gene expression is suppressed in male, as compared to female, blue gourami brains.

**Supplementary Materials:** The following supporting information can be downloaded at https://www.mdpi.com/article/10.3390/fishes9070287/s1: Supplementary File S1: Sequencing and transcriptome assembly stats; Supplementary File S2: DESeq2 stats; Supplementary File S3: DESeq2 results—brain, M vs. F; Supplementary File S4: g.profiler tables, up- and down-regulated in M vs. F; Supplementary File S5: primer sequences.

**Author Contributions:** Conceptualization, G.D. and A.M.; methodology, A.M.; formal analysis, A.M.; investigation, G.D. and A.M.; resources, G.D. and A.M.; data curation, A.M.; writing—original draft preparation, G.D.; writing—review and editing, G.D. and A.M.; visualization, G.D. and A.M.; project administration, A.M.; funding acquisition, G.D. and A.M. All authors have read and agreed to the published version of the manuscript.

**Funding:** This research was funded by an internal MIGAL grant.

**Institutional Review Board Statement:** This study was conducted in accordance with the Declaration of Helsinki and the AVMA Guidelines for the Euthanasia of Animals [23], under the supervision of an in-house veterinarian.

**Data Availability Statement:** Raw sequence reads are deposited in the SRA database (BioProject PRJNA1102721). The assembled transcriptome file is available upon request from the authors.

**Acknowledgments:** The authors thank Tali Shalit of The Crown Genomics institute of the Nancy and Stephen Grand Israel National Center for Personalized Medicine, Weizmann Institute of Science, Rehovot, Israel, for bioinformatics support.

**Conflicts of Interest:** The authors declare no conflicts of interest.

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
