# Peer review of "Sex Differences in the Brain Transcriptomes of Adult Blue Gourami Fish (Trichogaster trichopterus)"

_fishes, doi:10.3390/fishes9070287_

Round 1

Reviewer 1 Report (Previous Reviewer 2)

Comments and Suggestions for Authors In the resubmitted MS, the authors have added RT-qPCR experiments and improved the quality of this MS. The results are more reliable and logical. However, there are some minor points which need to be addressed. 1.When a gene name first appears, both the full name and the abbrivated form need to be given. Please check all the genes in the MS. According to the principle of zebrafish gene nomenclature, fish gene names should be written in lowercase and italicized form. 2.Generally, the regulation of steroid hormones produced in fish gonads on brain and pituitary act through negative feedback. Therefore, modification of Fig.5 is needed.   3. The format of references needs to be double checked. For example line 390.

Author Response

Comments 1: In the resubmitted MS, the authors have added RT-qPCR experiments and improved the quality of this MS. The results are more reliable and logical. However, there are some minor points which need to be addressed. 

1.When a gene name first appears, both the full name and the abbrivated form need to be given. Please check all the genes in the MS. According to the principle of zebrafish gene nomenclature, fish gene names should be written in lowercase and italicized form. 

Response 1: We have included the full names in the Abstract, Introduction and Results as requested. However, as our study is not concerned with zebrafish, zebrafish-specific gene nomenclature conventions need not apply.

Comments 2: 2.Generally, the regulation of steroid hormones produced in fish gonads on brain and pituitary act through negative feedback. Therefore, modification of Fig.5 is needed. 

Response 2: We have added feedback arrows from the gonads to the pituitary, in Fig. 5.

Comments 3: 3. The format of references needs to be double checked. For example line 390.

Response 3: Following the Reviewer’s comments, we have adjusted the reference formatting to the MDPI standard, and checked the reference list for consistency.

Reviewer 2 Report (New Reviewer)

Comments and Suggestions for Authors

This study identified the differentially expressed in male and female fish using high-throughput sequencing. However, the results are not well analyzed. I have specific major comments/questions I'd like the authors to address.

1. The abstract section should be focused and improve flow to improve readability.

2.The purpose of the study is not clear.

3.The section on test animals should include information on animal welfare and ethical approval.

4.Please add the validation experiment of RT-qPCR experiment for RNA-seq.

5. Line 210, why regardless of sex?

6. I want to know the relationship of brain, gonad, and dorsal fin in manuscript.

7. Materials and Methodneed revised more clear.

8. please add the section conclusion in the manuscript.

Comments on the Quality of English Language

The English of the manuscript is average, and the writing needs to be further improved

Author Response

Comments 1: The abstract section should be focused and improve flow to improve readability.

Response 1: The abstract was revised to shorten sentences and improve flow, where possible.

Comments 2: The purpose of the study is not clear.

Response 2: Following the Reviewer’s comments, the phrasing was modified in the Abstract (line 12) and Introduction (line 166) to emphasize the purpose of the study (namely, to assess sex differences in gene expression in non-gonadal tissues ostensibly affected by the sex-specific hormonal balance, e.g. the brain).

Comments 3: The section on test animals should include information on animal welfare and ethical approval.

Response 3: The section does state that “The study was conducted in accordance with the Declaration of Helsinki and the AVMA Guidelines for the Euthanasia of Animals [reference], under the supervision of an in-house veterinarian.” (lines 172-174). As there was no experimentation on the fish before they were euthanized at the fish farm, there was no experimental protocol to submit to the IRB. The authors have already discussed this topic at length with the Editorial Office after the original submission of the manuscript.

Comments 4: .Please add the validation experiment of RT-qPCR experiment for RNA-seq.

Response 4: The RT-qPCR validation experiment appears as section 2.7 in the Methods (line 273), section 3.4 in the Results (line 362) and Figure 4.

Comments 5: Line 210, why “regardless of sex”?

Response 5: This refers to all brain samples (from both males and females) clustering together in PCA when compared to other tissues (from females), indicating that the transcriptome divergence by tissue type is greater than by sex.

Comments 6: I want to know the relationship of brain, gonad, and dorsal fin in manuscript.

Response 6: The gonad samples (ovary, N=5) and dorsal fin samples (N=3) were taken from the same female BG as the brain samples, in order to serve as possible out-groups in clustering analysis of the brain samples. This explanation was added to the Methods (line 180-182).

Comments 7: “Materials and Method” need revised more clear.

Response 7: We have made several corrections in the Methods section; further input from the Reviewer on specific instances of lack of clarity could assist us in improving it.

Comments 8: please add the section “conclusion” in the manuscript.

Response 8: The “Conclusions” section appears as section 5 (line 761).

Reviewer 3 Report (New Reviewer)

Comments and Suggestions for Authors

This study aimed to investigate the gene expression differences in the brains of male and female Blue Gourami fish using RNA-seq analysis, with further validation by RT-qPCR. The manuscript characterizes differences in representative genes and pathways involved in growth and reproduction processes. While the study is generally well-conducted and foundational in its contribution, it is crucial that further investigation be conducted to comprehensively elucidate the biological significance of these observed sex differences.

Author Response

Comments: This study aimed to investigate the gene expression differences in the brains of male and female Blue Gourami fish using RNA-seq analysis, with further validation by RT-qPCR. The manuscript characterizes differences in representative genes and pathways involved in growth and reproduction processes. While the study is generally well-conducted and foundational in its contribution, it is crucial that further investigation be conducted to comprehensively elucidate the biological significance of these observed sex differences.

Response: We thank the Reviewer for the favorable assessment, and fully agree with the last statement. The direction and importance of future studies are emphasized in the last sentence of the Discussion.

Reviewer 4 Report (New Reviewer)

Comments and Suggestions for Authors

     Observation of differences in the transcriptomes of males and females is critical to understanding differences in the morphology, physiology and behavior between the sexes. Zebrafish has been the classical piscine model system, though science would do well to also consider other species. Degani and collaborators have done considerable work with blue gourami. In this manuscript, Degani and Meerson observed sex differences in brain transcriptomes of male and female blue gourami. The methods are appropriate, the execution sound, and the interpretations appropriate. Males showed down-regulation of expression of some brain function pathways. I have not substantive issues with the manuscript as presented, though there are many stylistic matters that must be attended to. I’ll discuss some of them below and also have marked the manuscript to aid in revision of presentation and prose.  

     The manuscript does not adhere well to journal stylistics in the following ways:

-         Citations should be numbered in the order in which they first appear in the text, and the References section should present them in that order.

-         In the citations themselves, the authors should capitalize the key words of article titles or not, but should be consistent. Journal names should be abbreviated and italicized, publication years should be in bold font, and volume number in italics.   

     Abstract. – Here and in several passages of the text, the authors write of “robust” suppression of neural function. I’m not sure what they mean by “robust” in this context. I wonder whether it is a bad translation from Hebrew. Would “broad” be better? Is any adjective needed? In any case, I suggest that a sentence at line 34 be recast as: … our results are novel in describing suppression of expression of genes related to neural function in adult BG males…

     Introduction. – The authors should mention the Latin names of all species at first usage, including zebrafish Danio rerio at line 41.

     At line 65 and throughout the manuscript, multiple species are referred to as fishES.

     At line 85, replace “BG” with “gouramis”; that is, researchers use COI as a marker for phylogenies across Anabantoidei.  

     Methods. – Supporting citations are need for all methods, including software packages used. We need supporting citations at lines 108, 135, 141, and 161.

     DE has to be spelled out at first usage at line 145.

     At line 162, the Raudven et al. reference does not appear in the References. I did not do a systematic check for all citations; there may be other such cases.

     At line 174, I am pretty sure that the authors should have written 60 seconds and 30 seconds, not minutes as written.

     Results. – We need supporting methods-related citations at lines 207 and 221.

     Wherever Supporting Information files are referenced, the authors should tell the reader which specific file to examine. Hence, specific files should be referred to at lines 210, 229, 239, 243, and 247.

     At line 213, the authors should not that the heatmap exhibits segregation of samples not only by sex, but also by tissue, an interesting key point missed. Of related note, Figure 2 at the lower right should have labels for the three dorsal fin and five ovary tissues, and the caption should mention these samples.    

     Figures 3 and 4 should be enlarged to make it more readable.

     Discussion. – At line 299, the authors write that sex-specific hormonal pathways are well known in other vertebrates. They should cite a well-chosen review here.

Comments on the Quality of English Language

There are some passages needing attention. I have marked the manuscript and will send the scan to the journal editorial office.

Author Response

Comments 1: -         Citations should be numbered in the order in which they first appear in the text, and the References section should present them in that order.

-         In the citations themselves, the authors should capitalize the key words of article titles or not, but should be consistent. Journal names should be abbreviated and italicized, publication years should be in bold font, and volume number in italics.  

Response 1: Following the Reviewer’s comments, we have adjusted the reference formatting to the MDPI standard as requested.

Comments 2:    Abstract. – Here and in several passages of the text, the authors write of “robust” suppression of neural function. I’m not sure what they mean by “robust” in this context. I wonder whether it is a bad translation from Hebrew. Would “broad” be better? Is any adjective needed? In any case, I suggest that a sentence at line 34 be recast as: … our results are novel in describing suppression of expression of genes related to neural function in adult BG males…

Response 2: In evolutionary biology, robustness of a biological system (also called biological or genetic robustness) is the persistence of a certain characteristic or trait in a system under perturbations or conditions of uncertainty. The choice of term was intentional, and the manuscript was written in English from the outset, and no part of it was translated from any other language. However, we don’t consider this adjective to be an essential point, and we have modified the phrasing (here and elsewhere) as per the Reviewer’s suggestion.

Comments 3:  Introduction. – The authors should mention the Latin names of all species at first usage, including zebrafish Danio rerio at line 41.

Response 3: Done (now appears in line 43)

Comments 4: At line 65 and throughout the manuscript, multiple species are referred to as fishES.

Response 4: Done as requested

Comments 5: At line 85, replace “BG” with “gouramis”; that is, researchers use COI as a marker for phylogenies across Anabantoidei. 

Response 5: “BG” was substituted with “gourami” throughout the manuscript.

Comments 6: Methods. – Supporting citations are need for all methods, including software packages used. We need supporting citations at lines 108, 135, 141, and 161.

Response 6: Done as requested, where possible. The first instance is a commercial package by Illumina, now stated as such.

Comments 7: DE has to be spelled out at first usage at line 145.

Response 7: Done as requested

Comments 8: At line 162, the Raudven et al. reference does not appear in the References. I did not do a systematic check for all citations; there may be other such cases.

Response 8: We thank the Reviewer for noticing this omission; it has now been corrected.

Comments 9: At line 174, I am pretty sure that the authors should have written 60 seconds and 30 seconds, not minutes as written.

Response 9: Actually, the values given are the correct times for the reverse transcription protocol used.

Comments 10: Results. – We need supporting methods-related citations at lines 207 and 221.

Response 10: Done as requested.

Comments 11: Wherever Supporting Information files are referenced, the authors should tell the reader which specific file to examine. Hence, specific files should be referred to at lines 210, 229, 239, 243, and 247.

Response 11: Done as requested.

Comments 12: At line 213, the authors should not that the heatmap exhibits segregation of samples not only by sex, but also by tissue, an interesting key point missed. Of related note, Figure 2 at the lower right should have labels for the three dorsal fin and five ovary tissues, and the caption should mention these samples.   

Response 12: Done as requested.

Comments 13: Figures 3 and 4 should be enlarged to make it more readable.

Response 13: Done as requested.

Comments 14: Discussion. – At line 299, the authors write that sex-specific hormonal pathways are well known in other vertebrates. They should cite a well-chosen review here.

Response 14: A citation of a chosen recent study was added (now at line 737).

Comments 15: Comments on the Quality of English Language

There are some passages needing attention. I have marked the manuscript and will send the scan to the journal editorial office. 

Response 15: We thank the Reviewer for the thorough proofreading of our manuscript, and have accepted almost all suggested changes in the revised version.

Round 2

Reviewer 2 Report (New Reviewer)

Comments and Suggestions for Authors

I suggest that the author make a comparison diagram between the high-throughput sequencing results and the RT-qPCR results in the manuscript.

Author Response

Comments: I suggest that the author make a comparison diagram between the high-throughput sequencing results and the RT-qPCR results in the manuscript.

Response: We have now included an additional chart (Figure 4B) that presents RNA-seq differential expression data for two of the receptors that were measured by RT-qPCR and appear in Figure 4A, namely PACAP-R1 and IGF1R. These were receptors that could be clearly identified in the RNA-seq and thus enabled a direct comparison to the RT-qPCR results.

This manuscript is a resubmission of an earlier submission. The following is a list of the peer review reports and author responses from that submission.

Round 1

Reviewer 1 Report

Comments and Suggestions for Authors

In this manuscript, transcriptome sequencing was conducted on female and male blue gourami (Trichogaster trichopterus) brain tissue, revealing 3568 differentially expressed genes. Among these, 1962 genes exhibited low expression in male brains, while 1606 genes showed high expression. However, the manuscript lacks in-depth data analysis and fails to present novel findings. Additionally, the writing quality is subpar, rendering it unsuitable for publication in Fishes.

Comments on the Quality of English Language

The English writing quality in this manuscript is subpar.

Reviewer 2 Report

Comments and Suggestions for Authors

In this MS, the authors carried out the brain RNA transcriptome sequencing and focused on the sex-biased gene expression. The finding is interesting and the MS is well written. However, there are several points which need to be addressed.  

1. The sexually different characteristics between female and male BG needs to be added in the background or discussion section to emphasize the significance of this study.

2. In this study, zebrafish is used for reference species, therefore, the percentage sequence mapped to the genome of zebrafish genome needs to be mentioned.

3.  Using RT-qPCR or other experiments by randomly selecting several genes, the authors needs to validate the quality and reliability of the sequencing data.

4. Are there any sex-biased expression of hormonal genes according to RNA-seq data? If so, their expression profile in both RNA-seq and PCR analysis are needed to support the conclusion of this MS.

5. In 233. What is BPG?